# CRISPR activation screen identifies BCL-2 proteins and B3GNT2 as drivers of cancer resistance to T cell-mediated cytotoxicity

Julia Joung [1,2,3,4,5,8 ✉], Paul C. Kirchgatterer[1,2,3,4,5], Ankita Singh[1,2,3,4,5], Jang H. Cho[1,2,3,4,5], Suchita P. Nety [1,2,3,4,5], Rebecca C. Larson[6,7], Rhiannon K. Macrae[1,2,3,4,5], Rebecca Deasy [2], Yuen-Yi Tseng[2], Marcela V. Maus [6,7] & Feng Zhang [1,2,3,4,5 ✉]

The cellular processes that govern tumor resistance to immunotherapy remain poorly understood. To gain insight into these processes, here we perform a genome-scale CRISPR activation screen for genes that enable human melanoma cells to evade cytotoxic T cell killing. Overexpression of four top candidate genes (*CD274* (PD-L1), *MCL1*, *JUNB*, and *B3GNT2*) conferred resistance in diverse cancer cell types and mouse xenografts. By investigating the resistance mechanisms, we find that MCL1 and JUNB modulate the mitochondrial apoptosis pathway. *JUNB* encodes a transcription factor that downregulates FasL and TRAIL receptors, upregulates the MCL1 relative BCL2A1, and activates the NF-κB pathway. *B3GNT2* encodes a poly-N-acetyllactosamine synthase that targets >10 ligands and receptors to disrupt interactions between tumor and T cells and reduce T cell activation. Inhibition of candidate genes sensitized tumor models to T cell cytotoxicity. Our results demonstrate that systematic gain-of-function screening can elucidate resistance pathways and identify potential targets for cancer immunotherapy.

[1] Department of Biological Engineering, MIT, Cambridge, MA 02139, USA. [2] Broad Institute of MIT and Harvard, Cambridge, MA 02142, USA. [3] Department of Brain and Cognitive Science, MIT, Cambridge, MA 02139, USA. [4] McGovern Institute for Brain Research at MIT, Cambridge, MA 02139, USA. [5] Howard Hughes Medical Institute, MIT, Cambridge, MA 02139, USA. [6] Department of Medicine, Massachusetts General Hospital and Harvard Medical School, Boston, MA, USA. [7] Cellular Immunotherapy Program, Cancer Center, Massachusetts General Hospital and Harvard Medical School, Boston, MA, USA. [8] Present address: Whitehead Institute, Cambridge, MA 02142, USA. ✉email: julia@joung.science; zhang@broadinstitute.org

By harnessing cytotoxic T cells of the immune system to eliminate cancer cells, cancer immunotherapy has transformed the foundation of cancer treatment and achieved notable clinical successes[1]. Nevertheless, resistance to immunotherapy is a major challenge[2–4], and elucidating the cellular pathways that confer resistance is critical for developing alternative and auxiliary strategies to expand the scope of immunotherapy. Small-scale studies have identified a small number of genes, including *CD274* (PD-L1), that enable tumors to evade the immune system, and PD-L1 inhibition in particular has been the focus of ongoing clinical development[4–9]. More recently, large-scale, loss-of-function genetic screens using CRISPR have identified additional genes that mediate resistance to T cell-induced cytotoxicity in the antigen presentation, interferon-γ (IFNγ)-sensing, tumor necrosis factor (TNF), and autophagy pathways[10–14]. However, in loss-of-function screens, candidate genes that can be inhibited to sensitize tumors against immunotherapy are depleted. As depletion screens have a lower dynamic range than enrichment screens[15], a more tractable approach is to perform a gain-of-function screen to enrich for genes that confer resistance upon upregulation[16] and could theoretically be inhibited to sensitize tumors against immunotherapy.

Here, we perform a genome-scale CRISPR activation (CRISPRa) screen for resistance against T cell cytotoxicity. Our screen identifies four candidate genes (*CD274* (PD-L1), *MCL1*, *JUNB*, and *B3GNT2*) that, upon upregulation, enable human melanoma cells to evade T cell killing in diverse cancer cell types and mouse xenografts. We elucidate the mechanisms of candidate genes and find that MCL1 and JUNB modulate the mitochondrial apoptosis pathway to promote resistance. *JUNB* encodes a transcription factor that downregulates FasL and TRAIL receptors, upregulates the MCL1 relative BCL2A1, and activates the NF-κB pathway. We find that *B3GNT2*, encoding a poly-N-acetyllactosamine synthase, operates in an orthogonal pathway to target >10 ligands and receptors to disrupt interactions between tumor and T cells and reduce T cell activation. Inhibition of candidate genes render tumor models more susceptible to T cell cytotoxicity. Together, our results show the feasibility of using systematic gain-of-function genetic screening to elucidate resistance pathways and identify potential therapeutic targets to expand the efficacy of cancer immunotherapy.

## Results

**CRISPR activation screen for T cell cytotoxicity resistance**. We first established a T cell cytotoxicity assay for measuring immunotherapy resistance. We transduced human primary CD4$^+$ and CD8$^+$ T cells with a T cell receptor (TCR) specific for the NY-ESO-1 antigen (NY-ESO-1:157-165 epitope) presented in an HLA-A*02-restricted manner (ESO T cells)[17]. When A375 (NY-ESO-1$^+$, HLA-A2$^+$) human melanoma cells were exposed to ESO T cells, we observed cytotoxicity that was specific to the presence of the NY-ESO-1 antigen and NY-ESO-1 TCR (Supplementary Fig. 1a–c). Cytotoxicity correlated with the effector to target (E:T) ratio (Supplementary Fig. 1b, c). We then transduced A375 cells with a genome-scale CRISPRa single-guide RNA (sgRNA) library consisting of 70,290 sgRNAs targeting every coding isoform from the RefSeq database (23,430 isoforms) to systematically identify genes that enable tumor cells to evade T cell killing upon upregulation[18] (Fig. 1a). We tested two T cell exposure strategies: acute (E:T ratio of 3 for 18 h) and chronic (E:T ratio of 2 for 3 days with three rounds of screening selection), in independent screens. We deep-sequenced the sgRNA library distribution in the surviving cells with or without ESO T cell exposure (Fig. 1a and Supplementary Fig. 1d–g). In the chronic exposure screen, we observed that the skew of the distribution increased after each round of screening selection (Supplementary Fig. 1e, g).

We performed MAGeCK[19] and FDR analyses to identify candidate genes that were enriched in cells cultured with ESO T cells relative to control (Fig. 1b, Supplementary Fig. 1h–k, and Supplementary Data 1–3). Both acute and chronic screening strategies exhibited high variability between replicates, as coculture screens, particularly those using primary cells from different donors, are often less well correlated than other types of screens. Indeed, comparable screens in a previous loss-of-function study[13] showed even higher variability (<10% overlap between top 1000 genes from two replicates compared to 30–60% overlap in our study; Supplementary Fig. 1h–k). Pathway analysis on 576 genes prioritized by MAGeCK (top 1% of multiple screening replicates combining the acute and chronic screens) revealed pathways were significantly enriched (FDR < 0.05) within these top candidates, including many that have been previously shown to be important for tumor immune evasion, such as lipopolysaccharide response, extrinsic apoptosis signaling, NF-κB activation, JAK-STAT signaling, antigen presentation, and Wnt signaling[10–14,20] (Fig. 1c and Supplementary Data 4). This analysis also highlighted pathways with previously underappreciated roles in regulating tumor response to T cell cytotoxicity, including glycosaminoglycan metabolism and carbohydrate catabolism, perhaps because we performed a gain-of-function, rather than loss-of-function, screen (Fig. 1c and Supplementary Data 4). Several candidate genes have been previously shown to mediate tumor immune evasion, such as *ATG3*[10], *DKK2*[21], *UHRF1*[22], and *CDYL*[22] (Supplementary Data 2), further indicating that our screen enriched for meaningful biological candidates.

To assess whether expression of the 576 candidate genes nominated by the screens was associated with local immune cytolytic activity in patient tumors (quantified using granzyme A and perforin 1 bulk transcriptome data)[23], we analyzed gene expression of 33 tumor types from The Cancer Genome Atlas (TCGA). We found that expression of 501 candidate genes positively correlated (FDR < 0.05) with cytolytic activity in at least 1 tumor type and 166 candidate genes were correlated across >25% of tumor types (Fig. 1d and Supplementary Fig. 2a, b). This is consistent with known immunotherapy resistance mediators, as high cytolytic activity selects for the emergence of evading tumor subclones (Supplementary Fig. 2c, d)[23]. Other candidate genes may have context-specific impacts across different types of cancers. We sought to evaluate whether the expression of candidate genes is associated with clinical outcome by analyzing 308 patient transcriptomes collected prior to immune checkpoint blockade therapy[24–29]. In this analysis, we found expression of candidate genes was significantly higher in nonresponders (Fig. 1e). MAGeCK analysis for genes that generally affect A375 cell fitness in the absence of ESO T cell co-culture showed that the MYC pathway governs fitness, with *MYC* and its antagonist *MXI1* as the top genes promoting or inhibiting cell fitness respectively (Supplementary Fig. 2e, f and Supplementary Data 5). Out of 576 candidate genes, 5 generally drive A375 cell fitness and 19 repress it (ranking in the top 1%; Supplementary Data 5).

**Validation of four top candidate genes**. To narrow our focus for further analysis, we selected the two most enriched genes from each screening strategy: *CD274* and *MCL1* from the acute screen, and *JUNB* and *B3GNT2* from the chronic exposure screen (Fig. 1b). Of these four candidates, *CD274* (PD-L1) is known to play a role in immune evasion, and it is currently the focus of immune checkpoint blockade therapies, supporting the design of our study[1]. We validated the four candidate genes by individually expressing three sgRNAs targeting each gene in A375 cells. For each candidate gene, at least two sgRNAs significantly increased survival against ESO T cells, verifying the screening results

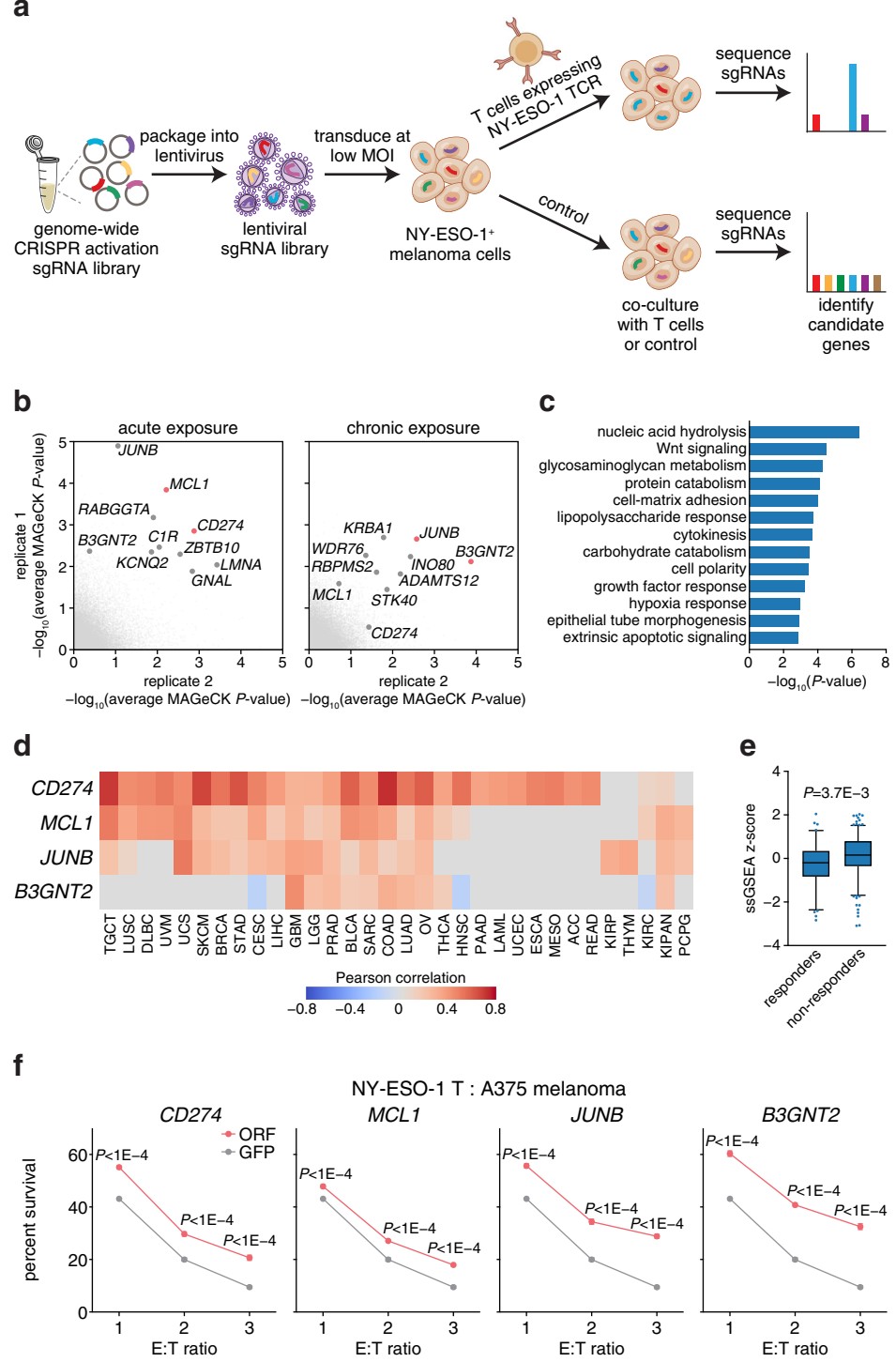

(*P* < 0.05; Supplementary Fig. 2g, h). For *JUNB* and *B3GNT2*, sgRNAs that produced higher target gene expression were more enriched in the screen and conferred more resistance, suggesting that resistance mediated by these genes depends on expression level (Supplementary Fig. 2g, h). Overexpression of ORFs encoding each of the four candidate genes increased survival against T cell cytotoxicity, excluding the possibility of potential CRISPRa off-target genes contributing to resistance (Fig. 1f).

We sought to further assess the clinical relevance of these top four candidate genes. By examining patient tumor samples from TCGA, we found that expression of *B3GNT2* was significantly higher than matched normal samples for 9 out of 31 types of cancer

(Supplementary Fig. 3a). Focal copy number gain of *MCL1* and *B3GNT2* occurred more frequently than losses across tumor types (in 95 and 81% of total cases respectively) (Supplementary Fig. 3b). Our TCGA analysis suggests that increases in both the expression of *B3GNT2* as well as copy number of *MCL1* and *B3GNT2* may generally promote tumor initiation or progression in the absence of immunotherapy. In melanoma patients treated with anti-PD-1 immunotherapy, higher *B3GNT2* expression was associated with poorer clinical response (Supplementary Fig. 3c, d)[27]. A different patient cohort showed that expression of *CD274* and *MCL1* significantly increased over the course of checkpoint blockade therapy in patients that did not respond to immunotherapy,

**Fig. 1 Genome-scale CRISPR activation screen identifies four candidate genes that confer resistance to T cell cytotoxicity. a** Schematic of the CRISPRa screen. NY-ESO-1[+] and HLA-A2[+] A375 melanoma cells were transduced with the pooled sgRNA library targeting more than 23,000 coding isoforms. A375 cells were exposed to primary CD4[+] and CD8[+] T cells expressing the T cell receptor (TCR) specific for the NY-ESO-1 antigen. Deep sequencing identified candidate genes. **b** Average MAGeCK analysis *P*-values for the acute and chronic exposure screens. Top candidate genes are annotated and the two most enriched genes from each screening strategy are highlighted in red. **c** Most significant pathways enriched among the 576 candidate genes. **d** Heatmap showing Pearson's correlation between expression of the top four candidate genes and cytolytic activity across patient tumors from TCGA. Only significant (FDR < 0.05) correlations are shown. **e** Box plots showing single-sample Gene Set Enrichment Analysis (ssGSEA)[60] of 576 candidate genes in 308 patient tumor samples[24–29]. Patient samples were collected prior to treatment with checkpoint inhibitors and classified as responders (*n* = 83) or nonresponders (*n* = 225) to immunotherapy. Box plots indicate median (middle line), 25th, 75th percentile (box), and 5th and 95th percentile (whiskers). Two-tailed *t* tests were performed. **f** Cell survival of A375 cells transduced with ORFs encoding candidate genes against ESO T cell cytotoxicity at different effector to target (E:T) ratios. Cell survival was measured using a luminescent cell viability assay and normalized to paired control cells that were not cultured with T cells. T cells were derived from three donors used in the CRISPRa screen, with *n* = 4 replicates per donor for *n* = 12 total. All values are mean ± s.e.m. Two-tailed *t* tests with adjustments for multiple comparisons were performed. Source data are provided in Source Data 1.

suggesting that upregulation of these genes may contribute to poor clinical outcome (Supplementary Fig. 3e)[29]. Further analysis of the Riaz et al. dataset showed expression of all four candidate genes significantly correlated with immune cytolytic activity, consistent with our analysis of TCGA data (Supplementary Fig. 3f). Together, these results suggest that increased expression of *MCL1* and *B3GNT2* may promote tumor progression and immune evasion in patients, whereas *JUNB* is not as clinically relevant across diverse tumor types.

**Candidate genes promote resistance in diverse contexts**. Next, we evaluated whether our screening results were generalizable to other contexts by testing different T cells and co-culture conditions. Overexpression of all candidate genes in A375 cells conferred resistance against ESO T cells from two additional donors that were not used in the CRISPRa screens (Supplementary Fig. 4a). We verified that candidate gene overexpression promoted resistance over time in an alternative T cell cytotoxicity assay based on secreted *Gaussia* luciferase (Supplementary Fig. 4b). In the absence of T cell cytotoxicity, upregulation of candidate genes did not consistently affect cell proliferation across the two cytotoxicity assays (Supplementary Fig. 4c, d). We investigated how expression level affects resistance by titrating the expression of candidate genes and found that expression correlated with resistance at lower levels of induction (extremely high expression levels of any protein, including GFP, reduced cell fitness, and sensitized cells to T cell cytotoxicity) (Supplementary Fig. 4e). The expression threshold above which we observed increased resistance corresponded to the baseline expression of 5–31% of cell lines from the Cancer Cell Line Encyclopedia[30], which suggested that the expression threshold for resistance is physiologically relevant (Supplementary Fig. 4f). We tested whether resistance conferred by candidate genes was specific to CD4[+] or CD8[+] T cells and found that candidate genes conferred resistance to both types of T cells (Supplementary Fig. 4g). To determine whether the resistance against T cells expressing the NY-ESO-1 TCR also applies to those expressing chimeric antigen receptors (CARs), we introduced candidate gene ORFs into A375 (AXL[+]) cells and co-cultured the cells with AXL-targeting CAR T cells[31]. Overexpression of each of the four candidate genes increased resistance against AXL-targeting CAR T cell cytotoxicity (Supplementary Fig. 4h). These results show that candidate genes confer resistance against different cytotoxic T cells and across co-culture conditions.

Next, we evaluated whether our screening results were generalizable to other types of cancers. We assayed candidate genes in seven additional cancer cell lines derived from five additional tissues [H1793 (NY-ESO-1[+], HLA-A2[−]) and H1299 (NY-ESO-1[+], HLA-A2[−]) non-small cell lung carcinomas, SW1417 (NY-ESO-1[−], HLA-A2[−]) colorectal adenocarcinoma,

OAW28 (NY-ESO-1[+], HLA-A2[−]) ovarian cystadenocarcinoma, A2058 (NY-ESO-1[−], HLA-A2[−]) melanoma, LN-18 (NY-ESO-1[+], HLA-A2[+]) glioblastoma, and SK-N-AS (NY-ESO-1[+], HLA-A2[−]) neuroblastoma]. Five of these cell lines expressed the NY-ESO-1 antigen endogenously, at varying levels (Supplementary Fig. 5a), and those that did not naturally express HLA-A2 or NY-ESO-1 were transduced with the appropriate expression vectors. We found that ORF overexpression of all four candidate genes significantly increased survival against T cell cytotoxicity in at least two additional cancer types (Fig. 2a, b and Supplementary Fig. 5b–h). Overexpression of *CD274* was not universally protective and did not confer resistance in cell lines with higher baseline expression, despite robust upregulation (Fig. 2b and Supplementary Fig. 5h). The effects of *MCL1* and *B3GNT2* overexpression could be generalized to six and seven of the additional cell lines respectively, demonstrating the broad applicability of these candidate genes to other cancer types and supporting the patient tumor analyses (Supplementary Fig. 3a, b).

To test the relevance of candidate genes for immunotherapy in vivo, we transduced A375 melanoma cells with dox-inducible candidate genes and subcutaneously engrafted these cells in immunocompromised NSG mice (Fig. 2c). At 2 days after subcutaneous tumor injection, we induced overexpression of candidate genes, and at 7 days we treated A375 xenografts with adoptive transfer of ESO T cells (Fig. 2c). In untreated control mice, we did not observe significant differences in tumor growth or host survival between the candidate genes and GFP control (Supplementary Fig. 5i, j). However, in mice treated with ESO T cells, overexpression of all four candidate genes significantly diminished the efficacy of adoptive cell transfer as measured by tumor growth and host survival (Fig. 2d, e). *JUNB* overexpression resulted in largely ineffective treatment, as *JUNB*-overexpressing xenografts displayed similar growth kinetics with and without ESO T cell treatment (Fig. 2d, e and Supplementary Fig. 5i, j).

**Mechanistic investigation of candidate genes**. We proceeded to investigate the mechanisms by which the candidate genes conferred resistance. As *CD274* has been extensively studied[1], we focused our mechanistic studies on the other three candidate genes. *MCL1* encodes a BCL-2 family protein that inhibits apoptosis by regulating mitochondrial outer membrane permeabilization, and *MCL1* overexpression is generally correlated with poor prognosis and resistance to most cancer therapeutics[32,33]. *JUNB* encodes a transcription factor that has been previously shown to downregulate an NKG2D ligand and mediate resistance against natural killer cells in mice[34]. *B3GNT2* encodes a beta-1,3-N-acetylglucosaminyltransferase. Although there are other B3GNT family members with overlapping functions, we did not identify any other B3GNT enzymes on our screen. This may be because B3GNT2 has the strongest poly-LacNAc synthesis

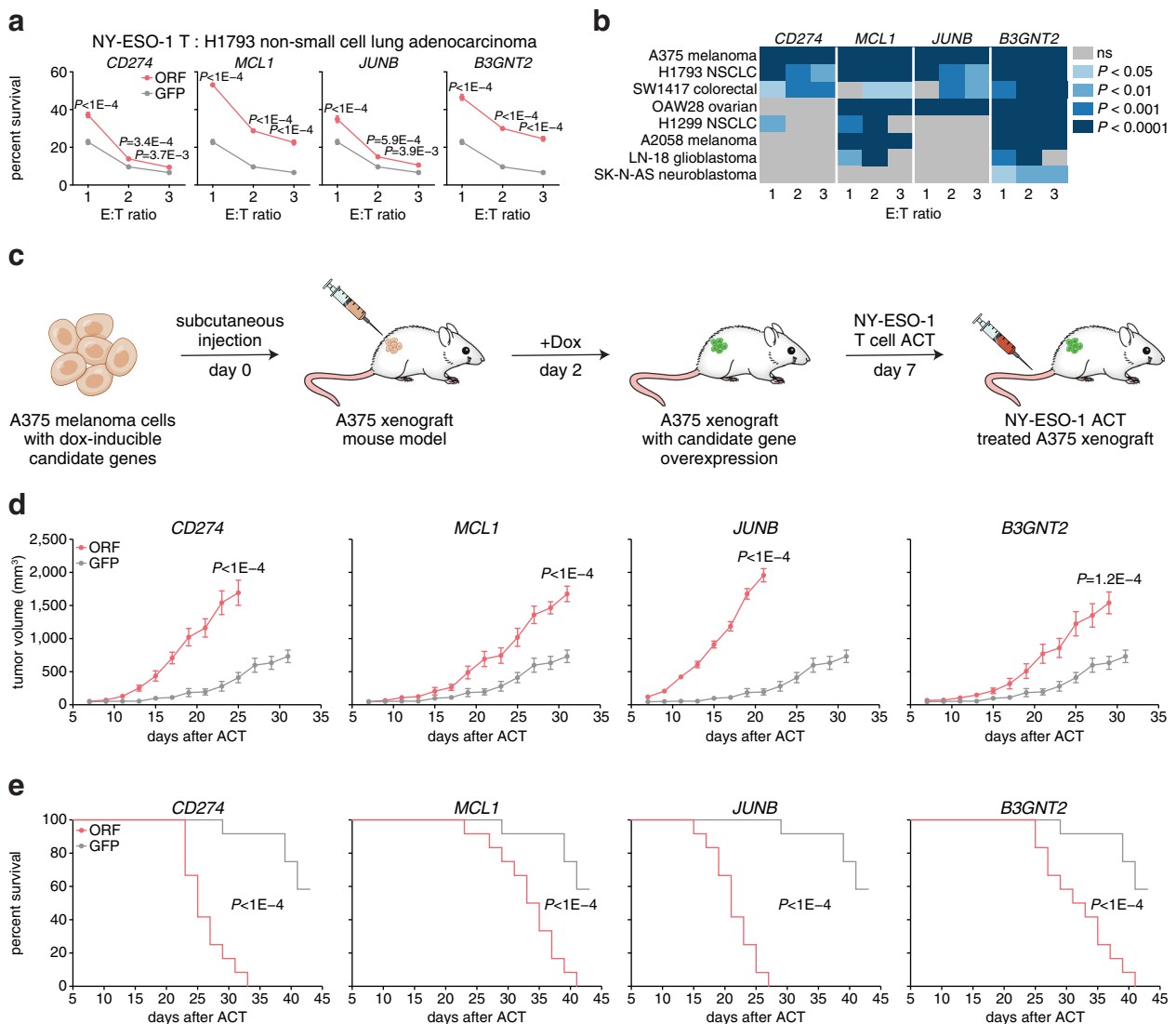

**Fig. 2 Candidate gene overexpression mediates resistance in other cell types and in vivo. a** Cell survival against ESO T cell cytotoxicity of H1793 (NY-ESO-1[+], HLA-A2[−]) non-small cell lung adenocarcinoma transduced with HLA-A2 and ORFs encoding candidate genes. $N = 12$. Two-tailed $t$ tests with adjustments for multiple comparisons were performed. **b** Heatmap summarizing results from ESO T cell cytotoxicity assays for eight cell lines derived from different tissues. Each value represents the significance of the difference between the survival of each ORF and GFP control. Two-tailed $t$ tests with adjustments for multiple comparisons were performed. **c** Schematic of the in vivo experiments to test the response of A375 xenografts overexpressing candidate genes to adoptive cell transfer (ACT) in NSG mice. **d, e** Tumor growth in mice receiving ACT of ESO T cells. Data are representative of two independent experiments. $N = 12$. **d** Tumor volume is shown. Two-tailed $t$ tests with adjustments for multiple comparisons were performed. **e** Overall survival is shown. Mantel–Cox log-rank tests were performed. All values are mean ± s.e.m. Source data are provided in Source Data 2.

activity in vitro relative to other B3GNT enzymes and is therefore considered the main poly-N-acetyllactosamine (poly-LacNAc) synthase[35,36]. In the immune system, *B3GNT2* is upregulated in T cells upon activation and *B3GNT2* knockout mice have lower poly-LacNAc on B and T cells, resulting in hyperactivity[35,37]. Single nucleotide polymorphisms that reduced expression of *B3GNT2* have been associated with autoimmune diseases[38–40]. To begin to understand the pathways related to each candidate gene, we performed RNA sequencing (RNA-seq) on A375 cells overexpressing each gene to characterize transcriptome changes. *JUNB* overexpression resulted in 632 differentially expressed genes with an absolute log fold change >1, compared to <15 genes for the other candidate genes, which is consistent with the role of *JUNB* in transcriptional regulation (Supplementary Fig. 6a and Supplementary Data 6). As the targets of *JUNB* and *B3GNT2* are relatively unknown compared to *MCL1*, we generated FLAG-

tagged ORFs of both genes for immunoprecipitation assays (Supplementary Fig. 6b). Chromatin immunoprecipitation sequencing (ChIP-seq) of JUNB and co-immunoprecipitation (co-IP) of B3GNT2 followed by mass spectrometry nominated 3517 and 414 targets, respectively (Supplementary Data 7 and 8).

To narrow down the possible pathways to those that affect tumor immune evasion, we assayed the effects of candidate gene overexpression on secretion and sensing of various cytokines involved in T cell cytotoxicity. We quantified IFNγ released by T cells in the cytotoxicity assay using ELISA and found that upregulation of *CD274* and *B3GNT2* reduced IFNγ secretion by T cells (Supplementary Fig. 6c). Overexpression of *CD274* reduced the tumor response to IFNγ, as indicated by reduced phosphorylation of STAT1, potentially resulting from a negative feedback mechanism (Supplementary Fig. 6d)[41]. We challenged A375 cells overexpressing each of the candidate genes with

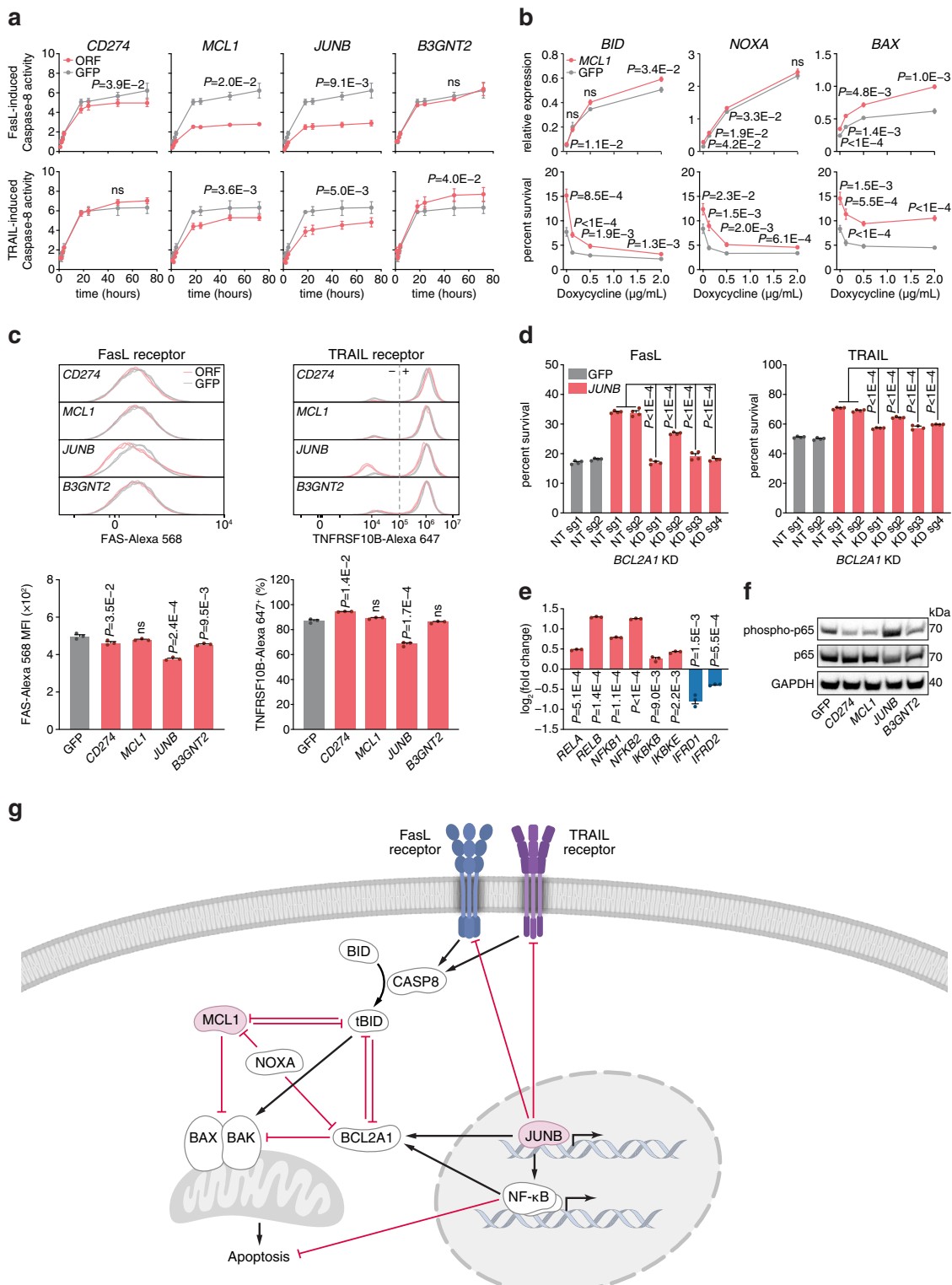

cytokines that mediate cytotoxicity, FasL, TRAIL, or TNF. We found that *MCL1* and *JUNB* overexpression significantly increased survival against FasL- and TRAIL-induced cell death, whereas *B3GNT2* overexpression reduced survival against TRAIL (Fig. 3a and Supplementary Fig. 7a).

**MCL1 and JUNB modulate the mitochondrial apoptosis pathway**. We examined components of the FasL and TRAIL

signaling pathways that could contribute to *MCL1*- and *JUNB*-mediated resistance. For MCL1, the potential interaction partners involved in FasL and TRAIL resistance have been identified in the previous studies[32]. We induced expression of these interaction partners in *MCL1*-overexpressing A375 cells and measured survival against T cell cytotoxicity. Induction of genes that more directly interact with *MCL1*, such as *BID*, *PMAIP1* (*NOXA*), and *BAX*, could offset resistance conferred by *MCL1* (Fig. 3b and Supplementary Fig. 7b). For *JUNB*, the extensive list of target

**Fig. 3 MCL1 and JUNB mediate resistance to FasL- and TRAIL-induced cell death through the mitochondrial apoptosis pathway. a** Caspase 8 activity measured using a colorimetric cleavage assay in A375 cells overexpressing candidate genes after treating with 500 ng/μL of FasL or TRAIL for 3 h. N = 3. Repeated measures ANOVA with adjustments for multiple comparisons were performed. **b** Dox-induction of genes in the mitochondrial apoptosis pathway in A375 cells overexpressing MCL1 or GFP. Cell survival against ESO T cell cytotoxicity (n = 8) and expression of MCL1 interaction partners (n = 4) were measured at different Dox concentrations. Two-tailed t tests with adjustments for multiple comparisons were performed. **c** Expression of cell surface FasL or TRAIL receptor, FAS or TNFRSF10B, measured by antibody staining and flow cytometry in A375 cells overexpressing candidate genes. Data are displayed as histograms (top), median fluorescence intensity (MFI; bottom left), and percent cells expressing receptor with gating at the gray dashed line (bottom right). N = 3. Two-tailed t tests were performed. **d** Cell survival against 500 ng/μL of FasL- or TRAIL-induced cell death in A375 cells overexpressing JUNB or GFP with BCL2A1 knocked down. N = 4. KD knockdown, NT non-targeting. Two-tailed t tests were performed. **e** Expression of the NF-κB pathway genes overlapping JUNB ChIP-seq and RNA-seq. Values represent fold change in A375 cells overexpressing JUNB relative to GFP control. N = 3. Two-tailed t tests with adjustments for multiple comparisons were performed. **f** Western blots of phosphorylated or total p65 (RELA) protein in A375 cells overexpressing candidate genes. Data are representative of two independent experiments. **g** Schematic describing the pathways MCL1 and JUNB are involved in to mediate resistance to T cell cytotoxicity. All values are mean ± s.e.m. ns not significant. Source data are provided in Source Data 3.

genes overlapping the ChIP- and RNA-seq datasets (Supplementary Data 7) suggests there could be multiple components involved, necessitating a more systematic approach. We first assayed cell surface expression of FasL and TRAIL receptors and found that JUNB overexpression moderately reduced expression of both receptors, FAS and TNFRSF10B, with TNFRSF10B expression reduced in 20% of the cells (Fig. 3c). We found that the JUNB target gene BCL2A1, which encodes an anti-apoptotic BCL-2 family protein that operates in parallel to MCL1[32], was upregulated 45-fold and included in the set of 576 candidate genes from the CRISPRa screen, suggesting that BCL2A1 may also contribute to FasL and TRAIL resistance (Supplementary Data 2 and 7). CRISPR inhibition (CRISPRi) knockdown of BCL2A1 in JUNB-overexpressing cells significantly decreased survival against cytotoxicity induced by FasL, TRAIL, and T cells (Fig. 3d and Supplementary Fig. 7c–e). In addition, JUNB alters expression of many different genes to activate the NF-κB pathway, including components of the NF-κB complex (RELA, RELB, NFKB1, and NFKB2), NF-κB activators (IKBKB and IKBKE), and NF-κB inhibitors (IFRD1)[42–44] (Fig. 3e). Perturbation of these genes by JUNB results in activation of the NF-κB pathway, as indicated by phosphorylation of p65 (RELA) (Fig. 3f). Similar to JUNB, NF-κB activation upregulates BCL2A1, thus creating a feed-forward loop for BCL2A1 upregulation[45,46]. Our results show that MCL1 and JUNB counteract FasL- and TRAIL-induced cell death by inhibiting the mitochondrial apoptosis pathway, further supporting the importance of the death receptor signaling pathway in immunotherapy[47,48] (Fig. 3g).

**B3GNT2 adds poly-LacNAc to >10 ligands and receptors.** Next, we turned to the resistance mechanism for B3GNT2. B3GNT2 overexpression in A375 cells increased intra- and extra-cellular poly-LacNAc as measured by tomato lectin staining (Supplementary Fig. 8a). As B3GNT2 adds poly-LacNAc to both N- and O-linked glycosylation[35], pretreating cells with either N- or O-linked glycosylation inhibitors, kifunensine or benzyl-2-acet-amido-2-deoxy-α-D-galactopyranoside (BAG) respectively, reduced poly-LacNAc added by B3GNT2 in a dosage-dependent manner (Supplementary Fig. 8a). As T cells that were co-cultured with B3GNT2-overexpressing A375 cells secreted less IFNγ (Supplementary Fig. 6c), we tested whether glycosylation inhibition could restore T cell activation. We found that pretreating B3GNT2-overexpressing A375 cells with both glycosylation inhibitors reversed the effects of B3GNT2 overexpression, resulting in increased T cell IFNγ secretion and reduced A375 survival, with kifunensine having a stronger effect (Fig. 4a and Supplementary Fig. 8b). As interaction between T cell and tumor cell surface ligands and receptors triggers IFNγ secretion, we assayed 21 ligands and receptors that were expressed in A375

for modifications by B3GNT2. We found that many of these proteins showed higher and broader ranges of molecular weights on western blots, potentially indicating increased presence of poly-LacNAc (Supplementary Fig. 8c, d). Enzymatic deglycosylation of the proteins confirmed that the increased molecular weights represented glycosylation, not other post-translational modifications (Supplementary Fig. 8d). Immunoprecipitation using tomato lectin or FLAG-tagged B3GNT2 further verified that B3GNT2 adds poly-LacNAc to ten ligands and receptors (CD276, CD70, CD58, NECTIN2, HLA-A, TNFRSF1A, IFNGR2, FAS, IFNAR1, MICB) (Fig. 4b and Supplementary Fig. 8e). At baseline, all ten ligands and receptors had some poly-LacNAc modifications, which increased in proportion and length upon overexpression of B3GNT2 (Fig. 4b). Pretreating A375 cells overexpressing B3GNT2 with either kifunensine or BAG showed that these ten ligands and receptors are primarily N-glycosylated, aligning with our finding that kifunensine treatment had a stronger effect on T cell IFNγ secretion and tumor cell survival (Fig. 4c). We found that these ten ligands and receptors had increased presence of poly-LacNAc at baseline in SW1417 colorectal adenocarcinoma cells, which express higher levels of endogenous B3GNT2 than A375 cells (Supplementary Fig. 8f).

We sought to determine whether increased poly-LacNAc on the B3GNT2 targets affected ligand–receptor interactions between tumor and T cells that facilitate T cell activation and subsequent cytotoxicity. By measuring binding of a panel of ten recombinant T cell proteins to A375 cells overexpressing B3GNT2, we found that binding of four T cell proteins [CD2, 4-1BB, TREML2 (TLT2), and NKG2D] was significantly reduced and binding of an antibody specific for HLA-A2:NY-ESO-1 was slightly reduced (Supplementary Fig. 9a). Treating B3GNT2-overexpressing A375 cells with kifunensine rescued the reduction in binding for all four T cell proteins, whereas BAG treatment only rescued NKG2D binding, consistent with our finding that most cell surface proteins targeted by B3GNT2 are N-glycosylated (Fig. 4c, d). In the case of HLA-A2:NY-ESO-1 antibody binding, kifunensine further reduced binding, potentially because antigen presentation was disrupted (Supplementary Fig. 9b). For T cell proteins CD2 and NKG2D as well as HLA-A2:NY-ESO-1 antibody, we have demonstrated that their known tumor interaction partners, CD58, MICB, and HLA-A respectively, are B3GNT2 target proteins[49,50] (Fig. 4c, d). However, for 4-1BB (TNFRSF9), the mechanism was not immediately clear because its known interaction partner, 4-1BBL (TNFSF9), was not modified by B3GNT2[51] (Supplementary Fig. 8c). CRISPR knockout of 4-1BBL resulted in a slight reduction of 4-1BB binding, potentially because of residual 4-1BBL protein (Supplementary Fig. 9c–e). These results suggest two possibilities: (1) increased poly-LacNAc on other cell surface ligands and receptors targeted by B3GNT2 disrupts the 4-1BB/4-1BBL binding; (2)

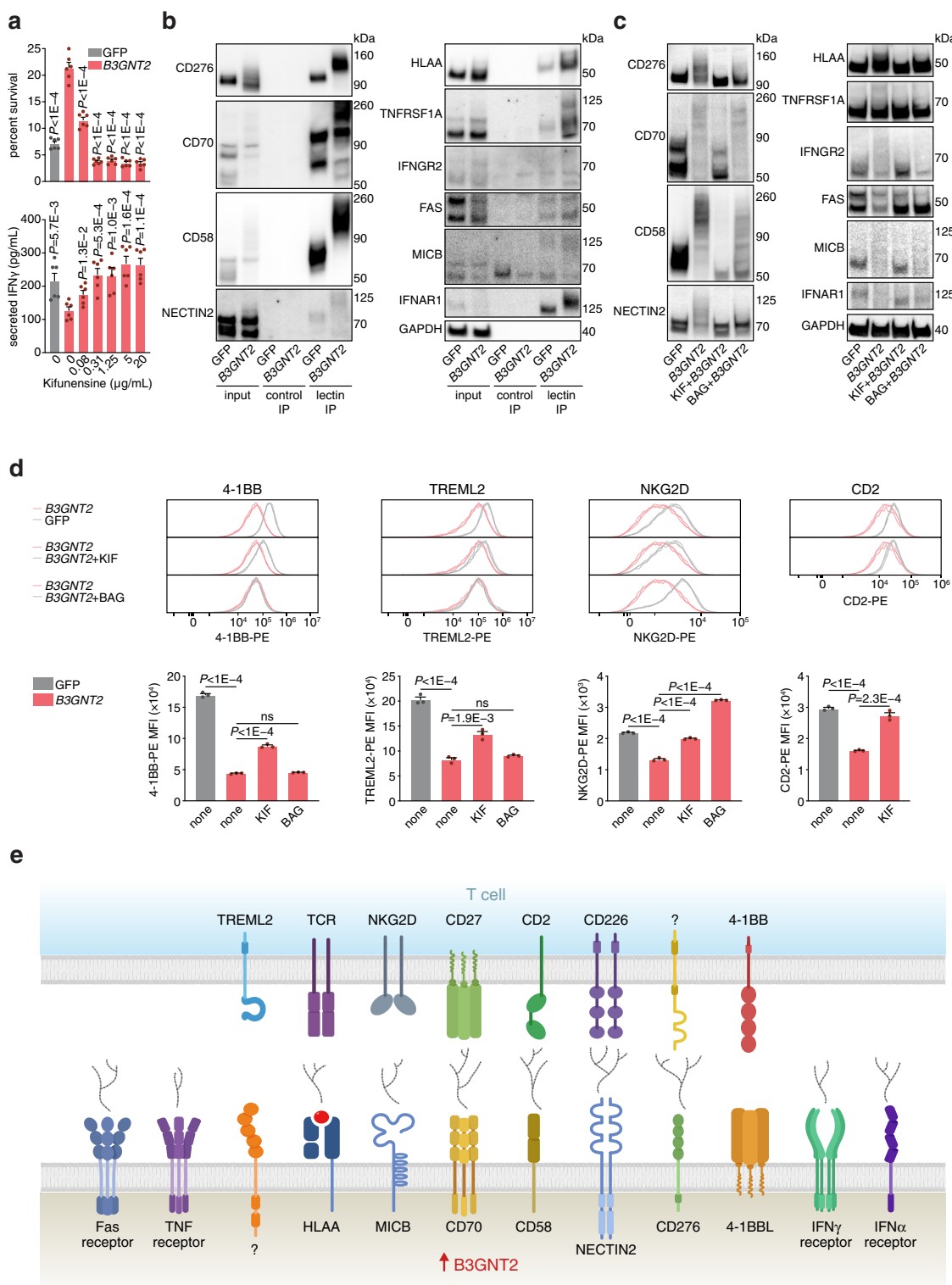

4-1BB binds to other unknown ligands that are targeted by B3GNT2. For TREML2, though some studies have suggested that TREML2 interacts with the B3GNT2 target protein CD276[52], CRISPR knockdown of CD276 did not affect binding to TREML2 (Supplementary Fig. 9f–h). Our finding aligns with a previous study showing that the interaction between TREML2 and CD276 does not occur in humans[53]. Taken together, we have shown that *B3GNT2* overexpression confers resistance against T cell cytotoxicity by adding poly-LacNAc on numerous proteins to interfere with ligand–receptor interactions between tumor and T cells, possibly by providing a survival advantage that outweighs increased sensitivity to TRAIL (Figs. 3a and 4e and Supplementary Fig. 7a).

**Candidate gene inhibition sensitizes tumors to immunotherapy.** To test whether inhibition of candidate genes could produce the opposite effect and render tumors more susceptible to T cell

**Fig. 4 B3GNT2 disrupts ligand–receptor interactions between tumor and T cells. a** Cell survival against T cell cytotoxicity (top) and T cell IFNγ secretion (bottom) in A375 cells overexpressing *B3GNT2* or GFP that have been treated with different concentrations of kifunensine. Kifunensine was used to pretreat A375 cells and was present during co-culture with T cells at E:T ratio of 3. Kifunensine-treated cells that were co-cultured with ESO T cells were compared to kifunensine-treated cells cultured in parallel without T cells to determine percent survival. $N = 6$. Two-tailed *t* tests were performed. **b** Tomato lectin IP of A375 cells overexpressing GFP or *B3GNT2* followed by western blot for different *B3GNT2* target proteins. 2% of the input and no lectin IP controls are shown. Data are representative of two independent experiments. **c** Western blots of A375 cells overexpressing *B3GNT2* or GFP that were treated with kifunensine (KIF) or benzyl-2-acetamido-2-deoxy-α-D-galactopyranoside (BAG) to remove *N*- or *O*-glycosylation respectively. Data are representative of two independent experiments. **d** Histograms (top) and corresponding median fluorescence intensity (MFI; bottom) showing binding of recombinant T cell proteins to A375 cells measured by flow cytometry. A375 cells were overexpressing GFP or *B3GNT2* and treated with KIF or BAG. $N = 3$. Two-tailed *t* tests were performed. **e** Schematic showing the tumor cell surface ligands and receptors modified by *B3GNT2* to disrupt interactions with T cells that mediate cytotoxicity. All values are mean ± s.e.m. ns not significant. Source data are provided in Source Data 4.

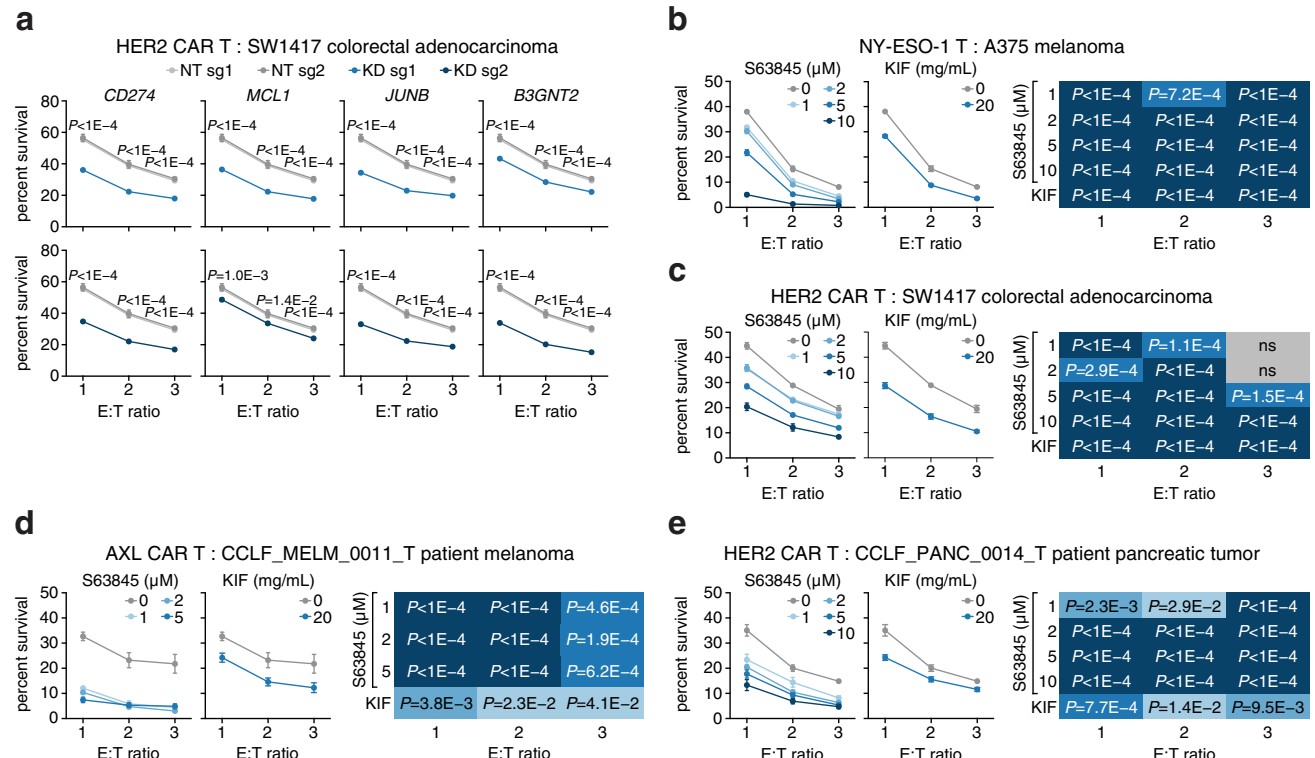

**Fig. 5 Inhibition of candidate genes sensitizes tumors to T cell cytotoxicity. a** Cell survival against HER2 CAR T cell cytotoxicity in SW1417 (HER2+) colorectal adenocarcinoma with different candidate genes knocked down using CRISPRi and 2 sgRNAs per gene. KD, knockdown. NT non-targeting. **b–e** Cell survival against T cell cytotoxicity in tumor cells treated with MCL1 or B3GNT2 small-molecule inhibitors, S63845 and kifunensine, respectively. Heatmaps show the significance of statistical analyses for each condition. **b** A375 (NY-ESO-1+, HLA-A2+) melanoma against ESO T cells. **c** SW1417 (HER2+) colorectal adenocarcinoma against HER2 CAR T cells. **d** CCLF_MELM_0011_T (AXL+) primary patient-derived melanoma model against AXL CAR T cells. **e** CCLF_PANC_0014_T (HER2+) primary patient-derived pancreatic adenocarcinoma against HER2 CAR T cells. All values are mean ± s.e.m with $n = 8$. ns not significant. Two-tailed *t* tests with adjustments for multiple comparisons were performed. Source data are provided in Source Data 5.

cytotoxicity, we designed CRISPR sgRNAs to knock down or knock out the four candidate genes and measured tumor survival against the T cell killing (Supplementary Fig. 10a, b). In SW1417 colorectal adenocarcinoma cells, which express relatively high levels of our candidate genes (Supplementary Fig. 5h), knockdown of all four candidate genes significantly decreased cell survival when cells were co-cultured with HER2 CAR or ESO T cells (Fig. 5a and Supplementary Fig. 10c). In A375 melanoma cells, knockdown of *CD274*, *MCL1*, and *JUNB* decreased cell survival, and in OAW28 ovarian cystadenocarcinoma cells, knockdown of *MCL1* and *JUNB* decreased cell survival (Supplementary Fig. 10d, e). Knockdown of *B3GNT2* did not affect survival in A375 and OAW28 cell lines, potentially because *B3GNT2* is expressed at relatively low levels in these cell lines (Supplementary Fig. 10a). We observed comparable results for

candidate gene knockout in SW1417 and A375 cells (Supplementary Fig. 10f–h). In addition to CRISPR perturbation, we tested the chemical inhibition of MCL1 and B3GNT2. Selective MCL1 inhibitors are already undergoing testing in clinical trials[33], and the dosage of these inhibitors could be adjusted to preferentially target MCL1-dependent tumor cells. The resistance mechanism of *B3GNT2* suggests that inhibition of extracellular poly-LacNAc could bolster immunotherapy. We therefore inhibited MCL1 using selective small-molecule inhibitors and B3GNT2 using kifunensine to generally inhibit *N*-linked glycosylation. Both chemical inhibition approaches reduced survival of A375 and SW1417 cells, as well as primary patient-derived melanoma and pancreatic adenocarcinoma models, against T cell cytotoxicity (Fig. 5b–e and Supplementary Fig. 10i, j). Our CRISPR and chemical inhibition results indicate that inhibition of

candidate genes in tumor cells enhances T cell killing and might be combined with current immunotherapy strategies to improve efficacy.

## Discussion

More generally, our results suggest that inhibition of B3GNT2 and BCL-2 family proteins, MCL1 and BCL2A1, could enhance the efficacy of immunotherapy and improve patient response. The high cross-validation rate of *MCL1* and *B3GNT2* across different cancer cell types and their frequency in patient tumor types suggest that the resistance effects are relatively cell-type independent. The distinct pathways of the candidate genes may have contributed to their respective differences in resistance to TCR and CAR T cell cytotoxicity. *MCL1* and *JUNB* over-expression may result in higher resistance against CAR-expressing T cell cytotoxicity because CAR-mediated killing may rely more on the mitochondrial apoptosis pathway for cytotoxicity[47,48]. By contrast, *B3GNT2* overexpression produces higher resistance against T cells expressing TCR than CAR because *B3GNT2* confers resistance by disrupting interactions between tumor and T cells to reduce T cell activation. As the CAR design includes multiple intracellular co-stimulatory domains that promote T cell activation[31], CAR function is not as affected by these disruptions. Characterizing resistance mechanisms thus might have the potential to inform the choice between TCR- and CAR-based immunotherapy.

We have shown here that genome-scale, gain-of-function genetic screens can discover genes involved in different biological processes that confer resistance to T cell cytotoxicity. We focused on the top four candidates and showed that overexpression of candidate genes conferred resistance in diverse types of cancers. Mechanistic investigation revealed that *MCL1* and *JUNB* over-expression modulate the mitochondrial apoptosis pathway to mediate resistance to FasL- and TRAIL-induced cell death. *JUNB* downregulates FasL and TRAIL receptors, upregulates *BCL2A1*, and activates the NF-κB pathway. *B3GNT2* promotes resistance through an orthogonal pathway by increasing poly-LacNAc on at least ten tumor ligands and receptors to reduce T cell activation, highlighting the importance of poly-LacNAc in immuno-oncology. Furthermore, inhibition of these genes sensitized both tumor cell lines and primary patient-derived tumor models to T cell killing. Our study complements results from previous loss-of-function screens and advances our understanding of the pathways that govern tumor immunotherapy. Moreover, our screening results serve as a starting point for further under-standing different pathways in tumor immune evasion and mechanistic studies to validate their roles. While additional stu-dies will need to be conducted to demonstrate that inhibition of other candidate genes found in this screen can enhance current approaches to immunotherapy, our results provide a strong foundation for such translational research.

## Methods

**Research compliance**. The designs of animal studies and procedures were approved by the Institutional Animal Care and Use Committee (IACUC) of the Broad Institute. Ethical compliance with IACUC protocols and institute standards was maintained. All human samples were obtained with informed consent and following institutional guidelines under protocols approved by the Institutional Review Board (IRB) at the Massachusetts General Hospital (MGH).

**Sequences and cloning**. The plasmids lenti dCAS-VP64_Blast (Addgene 61425), lenti sgRNA(MS2)_zeo backbone (Addgene 61427), and lentiMPHv2 (Addgene 89308) were used for CRISPR-Cas9 activation. The human SAM CRISPR activa-tion library (Addgene 1000000057) was used for CRISPR-Cas9 activation screen-ing. LentiCRISPRv2 (Addgene 52961) was used for CRISPR-Cas9 knockout. The Cas9 in lentiCRISPRv2 was replaced with dCas9-KRAB (Addgene 46911) and the Puromycin resistance gene was replaced with Blasticidin resistance gene (Addgene 75112) for CRISPR-Cas9 knockdown. Single-guide RNA (sgRNA) spacer

sequences used in this study are listed in Supplementary Table 1, and were cloned into the respective vectors[54]. The NY-ESO-1 T cell receptor (TCR) clone 1G4[17], AXL chimeric antigen receptor (CAR)[31], and HER2 CAR[31] were synthesized and cloned into the pHR TCR vector (Addgene 89347). The respective ORFs of can-didate genes [CD274 (NM_014143), MCL1 (NM_021960), JUNB (NM_002229), and B3GNT2 (NM_006577)] were synthesized and cloned into the plasmid pLX_TRC209 (Broad Genetic Perturbation Platform) for overexpression. HLA-A2 (Addgene 85162), ESO:HLA-A2, and *Gaussia* luciferase were cloned into pLX_TRC209 for stable expression. For dox-inducible upregulation, the EF1a promoter in pLX_TRC209 was replaced with the pTight promoter (Addgene 31877) and the plasmid pUltra-puro-RTTA3 (Addgene 58750) was used for rtTA.

**Cell culture**. HEK293FT cells (Thermo Fisher Scientific R70007) were maintained in high-glucose DMEM with GlutaMax and pyruvate (Thermo Fisher Scientific 10569010) supplemented with 10% fetal bovine serum (VWR 97068-085) and 1% penicillin/streptomycin (Thermo Fisher Scientific 15140122). Cells were passaged every other day at a ratio of 1:4 or 1:5 using TrypLE Express (Thermo Fisher Scientific 12604021).

All cancer cell lines [A375 melanoma (NY-ESO-1+, HLA-A2+; Millipore Sigma 88113005-1VL); H1793 non-small cell lung adenocarcinoma (NY-ESO-1+, HLA-A2−; ATCC CRL-5896), H1299 non-small cell lung carcinoma (NY-ESO-1+, HLA-A2−; ATCC CRL-5803), LN-18 glioblastoma (NY-ESO-1+, HLA-A2+; ATCC CRL-2610), SK-N-AS neuroblastoma (NY-ESO-1+, HLA-A2−; ATCC CRL-2137), A2058 melanoma (NY-ESO-1−, HLA-A2−; ATCC CRL-11147), OAW28 ovarian cystadenocarcinoma (NY-ESO-1+, HLA-A2−; Millipore Sigma 85101601-1VL), and SW1417 colorectal adenocarcinoma (NY-ESO-1−, HLA-A2−; ATCC CCL-238)] were maintained in RPMI 1640 with Glutamax (Thermo Fisher Scientific 61870127) supplemented with 10% fetal bovine serum and 1% penicillin/streptomycin. Cells were passaged every other day at a ratio of 1:3 to 1:6 using TrypLE Express.

Leukopaks from anonymous human healthy normal donors were purchased from the MGH blood bank under an IRB protocol of MGH. Leukopaks were processed using the Ficoll-based RosetteSep Human T Cell Enrichment Cocktail (StemCell Technologies 15061). Isolated CD4+ and CD8+ T cells were frozen in FBS with 10% DMSO with 20–50 × $10^6$ cells per vial. Once thawed, T cells were maintained in RPMI 1640 with Glutamax (Thermo Fisher Scientific 61870127) supplemented with 10% fetal bovine serum, 1% penicillin/streptomycin, and 20 IU/ mL IL-2 (Stemcell Technologies 78036.3). T cells were activated and expanded for 1 week using CD3/CD28 Dynabeads (Thermo Fisher Scientific 11132D). Beads were removed with two rounds of magnetic separation and T cells were frozen down (for in vitro cytotoxicity assays) or cultured for 1 week without beads (for adoptive cell transfer). CD4+ or CD8+ T cells were further purified using EasySep selection kits (StemCell Technologies 17852 and 17853, respectively) to assess the resistance of candidate genes against cytotoxicity produced from each T cell type. Experiments with T cells were performed using T cells derived from two to four unique donors with $n = 3$ or 4 biological replicates per donor.

**Lentivirus production and transduction**. One day prior to transfection, HEK293FT cells were seeded at ~40% confluency in T25, T75, or T225 flasks (Thermo Fisher Scientific 156367, 156499, or 159934). Cells were transfected the next day at ~90–99% confluency. For each T25 flask, 3.4 μg of the plasmid con-taining the vector of interest, 2.6 μg of psPAX2 (Addgene 12260), and 1.7 μg of pMD2.G (Addgene 12259) were transfected using 17.5 μL of Lipofectamine 3000 (Thermo Fisher Scientific L3000150), 15 μL of P3000 Enhancer (Thermo Fisher Scientific L3000150), and 1.25 mL of Opti-MEM (Thermo Fisher Scientific 31985070). Transfection parameters were scaled up linearly with flask area for T75 and T225 flasks. Media was changed 5 h after transfection. Virus supernatant was harvested 48 h post-transfection, filtered with a 0.45-μm PVDF filter (Millipore Sigma SLHV013SL), and concentrated when necessary via ultracentrifugation at 88,000 × $g$ for 2 h at 4 °C[54]. Virus supernatant was then aliquoted and stored at −80 °C.

Cancer cell lines were transduced by spinfection or mixing[54]. For spinfection, 3 × $10^6$ cells were seeded per well in a 12-well plate with 8 μg/mL Polybrene (Millipore Sigma TR-1003-G) and the appropriate volume in lentivirus. Cells were spinfected by centrifugation at 1000 × $g$ for 2 h at 33 °C. Cells were replated into T75 flasks with the appropriate antibiotic after 24 h. For mixing, 3 × $10^6$ cells were seeded in a T75 flask with 8 μg/mL Polybrene (Millipore Sigma TR-1003-G) and the appropriate volume in lentivirus. After 1 day, media was refreshed with the appropriate antibiotic, and cells were maintained under antibiotic selection for 5 days. Concentrations for selection agents were determined using a kill curve: 300 μg/mL Hygromycin (Thermo Fisher Scientific 10687010), 10 μg/mL Blasticidin (Thermo Fisher Scientific A1113903), 300 μg/mL Zeocin (Thermo Fisher Scientific R25001), and 1 μg/mL Puromycin (Thermo Fisher Scientific A1113803). T cells were transduced after 1 day of activation by mixing 1 × $10^6$ cells in 1 mL media with 8 μg/mL Polybrene and lentivirus in each well of a 24-well plate (Millipore Sigma CLS3527-100EA). The transduction efficiency of T cells was measured by sorting 1 × $10^6$ cells for GFP expression on the TCR vector after 7 days of activation. T cells used for experiments had transduction efficiencies of 80–90%.

**T cell cytotoxicity assays**. Expanded T cells were thawed and maintained in culture media for 8–10 h before incubation with cancer cells. Cancer cells were seeded in 96-well plates and allowed to attach for 3–4 h before T cells were added at the appropriate effector to target cell (E:T) ratio. Paired controls with no T cells added were included for each condition. After 18 h, cancer cells were washed twice with PBS to remove T cells, passaged, and cultured for 2 days. Primary patient-derived cell models were not passaged after T cell co-culture. Viability was measured using CellTiter-Glo (Promega G7571). For each E:T ratio, percent survival was calculated as the viability of the cells incubated with T cells divided by the viability of the paired control that was not incubated with T cells. For example, *CD274*-overexpressing melanoma cells that were co-cultured with ESO T cells were compared to *CD274*-overexpressing melanoma cells that were cultured without T cells in parallel. Cells treated with small-molecule inhibitors that were co-cultured with ESO T cells were compared to cells treated with small-molecule inhibitors cultured in parallel without T cells. As an alternative cytotoxicity assay, A375 cells stably expressing *Gaussia* luciferase were co-cultured with ESO T cells. At each time point, 10% of cell culture media was used for the *Gaussia* luciferase assay (Targeting Systems GAR-2B) to directly measure cytotoxicity.

**CRISPRa screen for resistance to T cell cytotoxicity**. A375 melanoma cells stably integrated with dCas9-VP64 (Addgene 61425) and MS2-P65-HSF1 (Addgene 61426) were transduced with the pooled CRISPRa sgRNA library (Addgene 1000000057) as described above at an MOI of 0.3, with a minimal representation of 500 transduced cells per sgRNA in each replicate. For the acute exposure screen, A375 cells were co-cultured with T cells expressing the NY-ESO-1 TCR, unmodified T cells, or no T cells at E:T ratio of 3. Each screen contained two replicates with T cells from different donors. After 18 h of co-culture, cells were washed twice with PBS to remove T cells, passaged, and cultured for 2 days before genomic DNA was harvested. For the chronic exposure screen, A375 cells were co-cultured with T cells expressing the NY-ESO-1 TCR or no T cells at E:T ratio of 2. Screening replicates used T cells from the same donor and each round of screening selection used T cells from different donors. After 3 days of co-culture, cells were washed twice with PBS to remove T cells, passaged, and cultured for 2 days before seeding for the next round of screening selection. After 3 rounds of screening selection, genomic DNA was harvested. MAGeCK RRA analysis[19] was used to analyze the screens and identify candidate genes. A set of 576 candidate genes that ranked in the top 1% and overlapped at least two screening replicates (combining the acute and chronic exposure screens) were used for pathway and cytolytic activity analyses. The FDR of screening results was estimated using a set of 311 negative control housekeeping genes consisting of ribosomal proteins, RNA polymerases, translation factors, mitochondrial ribosomal proteins, *GAPDH*, and *ACTB* (Supplementary Data 2). For each screening replicate, the FDR of each candidate gene was measured as the fraction of negative control genes with higher average sgRNA enrichment than the candidate gene. To validate the top four candidate genes from the screens, sgRNAs targeting candidate genes from the genome-scale library were individually cloned and transduced into A375 cells. Validation was performed using T cell cytotoxic assays at an E:T ratio of 3 as described above.

**Pathway enrichment analysis**. Pathway enrichment analysis of the top 576 candidate genes was performed using g:Profiler[55]. GO:BP pathways with between 5 and 200 genes that were significantly enriched (*FDR* < 0.05) were included. To identify nonoverlapping pathways, the enriched pathways were sorted by *FDR* and any pathway that had more than 30% genes overlapping a different pathway with lower FDR was excluded.

**The Cancer Genome Atlas (TCGA) analysis**. TCGA copy number variation and RNA-seq data were downloaded from the Firehose Broad GDAC (http://gdac.broadinstitute.org/) using the TCGA2STAT package for R[56]. The RNA-seq data was normalized using RSEM and log2 transformed. Local tumor immune cytolytic activity was determined as the geometric mean of granzyme A (*GZMA*) and perforin 1 (*PRF1*) RNA-seq expression[13,23]. For each gene in the TCGA RNA-seq dataset, Pearson's correlation between cytolytic activity and expression was calculated. Significance was evaluated using Fisher transformation of Pearson's correlation followed by Benjamini–Hochberg procedure to determine the *FDR*. For visualization, heatmaps with hierarchical clustering using Ward's linkage were generated using Python's Seaborn clustermap (https://github.com/mwaskom/seaborn/).

For the prevalence of increased expression and copy number of the top four candidate genes, TCGA RNA-seq data (https://www.cancer.gov/tcga) was analyzed using GEPIA[57]. TCGA tumor samples were matched with TCGA normal and GTEx data and filtered for $|\log_2(fold\ change)| \geq 1$. Genes were considered significantly differentially expressed if the *P* value was greater than 0.05 *FDR* correction. Copy number variation was reported using the NCI Genomic Data Commons[58].

**Single-sample gene set enrichment analysis (ssGSEA)**. A total of 308 unique patient tumor transcriptomes that were collected prior to immunotherapy were used for ssGSEA[24–29]. As processed data was not available for the Gide et al.

dataset[26], fastq files were downloaded and expression levels were estimated using RSEM v1.3.1[59] as described below. Expression values for replicates from the same patient were averaged. ssGSEA[60] as implemented by GSEAPY v0.10.4 was performed on each sample using default parameters to determine the normalized enrichment score of the 576 candidate genes. The z-score of the normalized enrichment scores was calculated on each dataset and aggregated. Patients were classified as responders (i.e., RECIST categories of complete response or partial response, clinical benefit, and no tumor progression) or nonresponders (i.e., RECIST categories of stable disease or progressive disease, no clinical benefit, and tumor progression) based on the reported response to subsequent anti-PD-1 or anti-CTLA-4 checkpoint blockade therapy.

**Indel analysis**. Cells plated in 96-well plates were grown to 60–80% confluency and assessed for indel rates[54]. Genomic DNA was harvested from cells using Quick-Extract DNA Extraction kit (Lucigen QE09050). The genomic region flanking the site of interest was amplified using NEBNext High Fidelity 2× PCR Master Mix (New England BioLabs M0541L), first with region-specific primers (Supplementary Table 2) for 15 cycles and followed by barcoded primers for 15 cycles. PCR products were sequenced on the Illumina MiSeq platform (>10,000 reads per condition), and indel rates were determined using a published Python script[54].

**qPCR quantification of transcript expression**. Cells were seeded in 96-well plates and grown to 60–90% confluency prior to RT-qPCR[54]. Cells were lysed by adding Lysis Buffer [4.8 mM Tris pH 8.0 (Thermo Fisher Scientific AM9855G), 4.8 mM Tris pH 7.5 (Thermo Fisher Scientific 15567027), 0.5 mM MgCl₂ (Thermo Fisher Scientific AM9530G), 0.44 mM CaCl₂ (Millipore Sigma 21115), 10 μM DTT (Promega P1171), 0.1% wt/vol Triton X-114 (Millipore Sigma X-114), 6 U/mL Proteinase K (Millipore Sigma P2308), 300 U/mL DNAse I (Millipore Sigma D2821)] and mixing. After 6–12 min at room temperature, lysis was terminated by adding Stop Lysis Buffer [1 mM Proteinase K inhibitor (Millipore Sigma 539470), 90 mM EGTA (Millipore Sigma E3889), 113 μM DTT (Promega P1171)] and mixing. RNA in cell lysate was reverse transcribed into cDNA using the RevertAid RT Reverse Transcription Kit (Thermo Fisher Scientific K1691) with 3.5 μM oligo dT (Integrated DNA Technologies; TTTTTTTTTTTTTTTTTTTTTTNN). The reverse transcription reaction was run with the following cycle conditions: 25 °C for 10 min, 37 °C for 1 h, and 95 °C for 5 min. TaqMan qPCR was performed on the cDNA using TaqMan Fast Advanced Master Mix (Thermo Fisher Scientific 4444557) with custom [Integrated DNA Technologies; *B3GNT2*-Fwd (GGGCAGGCTCTCCAATATAAG), *B3GNT2*-probe (/56-FAM/TGAACTACT/Zen/GCGAACCTGACCTGA/3IABkFQ/), *B3GNT2*-Rev (GGCATCTCAAATA-CAGCAGAAAG)] or readymade probes [Thermo Fisher Scientific; *CD274* (Hs00204257_m1), *MCL1* (Hs01050896_m1), *JUNB* (Hs00357891_s1), *BID* (Hs00609632_m1), *PMAIP1* (Hs00560402_m1), *BBC3* (Hs00248075_m1), *BAD* (Hs00188930_m1), *BAX* (Hs00180269_m1), *BAK1* (Hs00832876_g1), *BCL2A1* (Hs06637394_s1), *CD276* (Hs00987207_m1)].

**Adoptive cell transfer and in vivo validation**. Specific pathogen-free facilities at the Broad Institute was used for the storage and care of all mice. Mice were housed at a temperature of 67–73 °F, relative humidity of 30–60%, and maintained in a 12 h light–dark cycle. Female NSG mice (strain 005557) aged 4–6 weeks were purchased from The Jackson Laboratory and used for tumor induction experiments. A375 cells were transduced with dox-inducible candidate genes. NSG mice were subcutaneously injected with $1 \times 10^6$ A375 cells. After 2 days of tumor xenograft implantation, mice were switched to 1000 mg/kg doxycycline diet (Envigo TD.05298). At 7 days after tumor implantation, for the adoptive cell transfer conditions, $2 \times 10^7$ ESO T cells were intravenously injected in a blinded manner. Each tumor was measured every 2 days beginning on day 7 after ACT until the survival endpoint was reached. Measurements were assessed manually using the longest dimension (length) and the longest perpendicular dimension (width). Tumor volume was estimated with the formula: $(L \times W^2)/2$. The maximal tumor size permitted by the IACUC of the Broad Institute was 2000 mm³. Mice with tumor volumes greater than 2000 mm³ were euthanized. CO₂ inhalation was used to euthanize mice. No statistical methods were used to predetermine sample size. The sample size was determined based on prior knowledge of the variability of experiments with ACT. Animals were randomized before treatment and no blinding was performed for tumor measurements.

**Bulk RNA sequencing and data analysis**. RNA from cells plated in 24-well plates and grown to 60–90% confluency was harvested using the RNeasy Plus Mini Kit (Qiagen 74134). RNA-seq libraries were prepared using NEBNext Ultra RNA Library Prep Kit for Illumina (New England Biolabs E7530S) and deep-sequenced on the Illumina NextSeq platform (>9 million reads per biological replicate). Bowtie[61] index was created based on the human hg38 UCSC genome and RefSeq transcriptome. Next, RSEM v1.3.1[59] was run with command-line options "--estimate-rspd --bowtie-chunkmbs 512 --paired-end" to align paired-end reads directly to this index using Bowtie and estimate expression levels in transcripts per million (TPM) based on the alignments.

To identify genes that were differentially expressed as a result of ORF overexpression, RSEM's TPM estimates for each transcript were transformed to

log-space by taking $\log_2(\text{TPM} + 1)$. Transcripts were considered detected if their expression level was equal to or above 10. All genes detected in at least three libraries were used to find differentially expressed genes. The Student's *t* test was performed on the ORF overexpression condition against GFP control condition. Only genes that were significant (*P* value pass 0.01 *FDR* correction) were reported.

**Chromatin immunoprecipitation with sequencing (ChIP-seq)**. Cells were plated in 10-cm cell culture dishes and grown to 60–80% confluency. For each condition, two biological replicates were harvested for ChIP-seq. Formaldehyde (Millipore Sigma 252549) was added directly to the growth media for a final concentration of 1% and cells were incubated at 37 °C for 10 min to initiate chromatin fixation. Fixation was quenched by adding 2.5 M glycine (Millipore Sigma G7126) in PBS for a final concentration of 125 mM glycine and incubated at room temperature for 5 min. Cells were then washed with ice-cold PBS, scraped, and pelleted at $1000 \times g$ for 5 min.

Cell pellets were prepared for ChIP-seq using the Epigenomics Alternative Mag Bead ChIP Protocol v2.0[62] Briefly, cell pellets were resuspended in 100 μL of lysis buffer (1% SDS, 10 mM EDTA, 50 mM Tris-HCL pH 8.1) containing protease inhibitor cocktail (Millipore Sigma 05892791001) and incubated for 10 min at 4 °C. Then 400 μL of dilution buffer (0.01% SDS, 1.1% Triton X-100, 1.2 mM EDTA, 16.7 mM Tris-HCl pH 8.1, and 167 mM NaCl) containing protease inhibitor cocktail was added. Samples were pulse sonicated with two rounds of 10 min (30 s on-off cycles, high frequency) in a rotating water bath sonicator (Diagenode Bioruptor) with 5 min on ice between each round. 10 μL of sonicated sample was set aside as input control. Then 500 μL of dilution buffer (0.01% SDS, 1.1% Triton X-100, 1.2 mM EDTA, 16.7 mM Tris-HCl pH 8.1, and 167 mM NaCl) containing protease inhibitor cocktail and 1 μL of anti-FLAG (Millipore Sigma F3165-1MG) was added to the sonicated sample. ChIP samples were rotated end-over-end overnight at 4 °C.

For each ChIP, 50 μL of Protein A/G Magnetic Beads (Thermo Fisher Scientific 88802) was washed with 1 mL of blocking buffer (0.5% TWEEN and 0.5% BSA in PBS) containing protease inhibitor cocktail twice before resuspending in 100 μL of blocking buffer. ChIP samples were transferred to the beads and rotated end-over-end for 1 h at 4 °C. ChIP supernatant was then removed and the beads were washed twice with 200 μL of RIPA low salt buffer (0.1% SDS, 1% Triton x-100, 1 mM EDTA, 20 mM Tris-HCl pH 8.1, 140 mM NaCl, 0.1% DOC), twice with 200 μL of RIPA high salt buffer (0.1% SDS, 1% Triton x-100, 1 mM EDTA, 20 mM Tris-HCl pH 8.1, 500 mM NaCl, 0.1% DOC), twice with 200 μL of LiCl wash buffer (250 mM LiCl, 1% NP40, 1% DOC, 1 mM EDTA,10 mM Tris-HCl pH 8.1), and twice with 200 μL of TE (10 mM Tris-HCl pH 8.0, 1 mM EDTA pH 8.0). ChIP samples were eluted with 50 μL of elution buffer (10 mM Tris-HCl pH 8.0, 5 mM EDTA, 300 mM NaCl, 0.1% SDS). In total, 40 μL of water was added to the input control samples. 8 μL of reverse cross-linking buffer (250 mM Tris-HCl pH 6.5, 62.5 mM EDTA pH 8.0, 1.25 M NaCl, 5 mg/ml Proteinase K, 62.5 μg/ml RNAse A) was added to the ChIP and input control samples and then incubated at 65 °C for 5 h. After reverse cross-linking, samples were purified using 116 μL of SPRIselect Reagent (Beckman Coulter B23318).

ChIP samples were prepared for NGS with NEBNext Ultra II DNA Library Prep Kit for Illumina (New England Biolabs E7645S) and deep-sequenced on the Illumina NextSeq platform (>60 million reads per condition). Bowtie 1.2.3[61] was used to align paired-end reads to the human hg38 UCSC genome with command-line options q -X 300 --sam --chunkmbs 512". Next, biological replicates were merged and Model-based Analysis of ChIP-seq (MACS)[63] was run with command-line options "-g hs -B -S --mfold 6,30" to identify TF peaks. HOMER[64] was used to discover motifs in the TF peak regions identified by MACS. TFs were considered potential regulators of a candidate gene if the TF peak region identified by MACS overlapped with the promoter region of the candidate gene. Promoter regions were defined as 2000 bp upstream and 500 bp downstream of RefSeq transcriptional start sites.

**Co-immunoprecipitation (co-IP) and mass spectrometry**. Cells were plated in 10-cm cell culture dishes and grown to 60–80% confluency. For each condition, two biological replicates were harvested for co-IP. Cells were washed with PBS and 4 mL of lysis buffer (20 mM HEPES, 1% Triton X-100, 150 mM NaCl, 1 mM EDTA, and 10% glycerol) containing protease inhibitor cocktail was added. Cells were scraped, and the lysate was incubated at 4 °C under rotary agitation for 1 h. The lysate was centrifuged at $14,000 \times g$ for 10 min at 4 °C. The supernatant was transferred to a new tube, and an aliquot was taken as the input. The remaining lysate was split into two tubes for the FLAG and IgG control conditions. For mass spectrometry, 10 μg/mL mouse Anti-FLAG (Millipore Sigma F3165-1MG) and IgG control (Millipore Sigma 12-371) were added to the respective conditions. For tomato lectin IP western blots, 20 μg/mL biotinylated tomato lectin (Vector Laboratories B1175) was added. For co-IP western blots, 10 μg/mL chicken anti-FLAG (Aves labs ET-DY100) and IgY control (R&D Systems AB-101-C) antibodies were biotinylated (Thermo Fisher Scientific 90407) and added to the respective conditions. Lysates with antibodies were incubated at 4 °C under rotary agitation overnight. For each mL of lysate, 50 μL of Pierce Protein A/G Magnetic Beads (Mass spectrometry; Thermo Fisher Scientific 88803) or Pierce Streptavidin Magnetic Beads (Western blot; Thermo Fisher Scientific 88817) was washed twice with lysis buffer. Lysates with antibodies were added to the beads and incubated at

4 °C under rotary agitation for 4 h. Beads were washed with lysis buffer three times and resuspended in lysis buffer for storage.

Magnetic beads were resuspended in 100 mM Tris pH 7.8, reduced, alkylated, and digested with trypsin at 37 °C overnight. This solution was subjected to solid-phase extraction to concentrate the peptides and remove unwanted reagents followed by injection onto a Shimadzu HPLC with a fraction collector. Eight fractions were collected, and after concentration, were injected on a Waters NanoAcquity HPLC equipped with a self-packed Aeris 3-μm C18 analytical column 0.075 mm by 20 cm (Phenomenex). Peptides were eluted using standard reverse-phase gradients. The effluent from the column was analyzed using a Thermo Orbitrap Elite mass spectrometer (nanospray configuration) operated in a data-dependent manner for 54 min. The resulting fragmentation spectra were correlated against the known database using Proteome Discover 1.4 (Thermo Fisher Scientific). Scaffold Q + S (Proteome Software) was used to provide consensus reports for the identified proteins.

**Cytokine assays**. To challenge cells with cytokines, cells were incubated with Interferon-γ (IFNγ; Cell Signaling Technology 80385S), FasL (AdipoGen AG-40B-0130-3010), TRAIL (R&D Systems 375-TL-010), or TNF-α (AdipoGen AG-40B-0019-3010) for 24 h. TRAIL was crosslinked by incubating with anti-His Tag antibody (Thermo Fisher Scientific MA121315, 1:500) for 15 min at room temperature. Cell viability was measured using CellTiter-Glo (Promega G7571) and protein was harvested for western blots. For evaluating Caspase 8 activity, cells were incubated with FasL or crosslinked TRAIL for 3 h and harvested for Caspase 8 colorimetric assay (R&D Systems K113-100). IFNγ in the cell culture media of the T cell cytotoxic assay was quantified using an ELISA kit (Thermo Fisher Scientific KHC4021).

**Small-molecule inhibition**. For glycosylation inhibition, cells were pretreated with 20 μg/mL Kifunensine (Cayman Chemical 10009437) or 2 mM Benzyl-2-acetamido-2-deoxy-alpha-D-galactopyranoside (BAG; Millipore Sigma B4894-100MG) for 48 h to inhibit N- and O-glycosylation, respectively unless otherwise indicated before incubation with T cells. For *MCL1* inhibition, cells were pretreated with 1–10 μM of S63845 (Selleck Chemicals S8383) or AZD5991 (Selleck Chemicals S8643) for 4 h before incubation with T cells. Both glycosylation and *MCL1* inhibitors were maintained at indicated concentrations during co-culture with T cells.

**Western blot**. Protein lysates were harvested with RIPA lysis buffer (Cell Signaling Technologies 9806S) containing protease inhibitor cocktail (Millipore Sigma 05892791001). Samples were standardized for protein concentration using the Pierce BCA protein assay (VWR 23227), and incubated at 70 °C for 10 min under reducing conditions. To determine the presence of glycosylation, samples were treated with Protein Deglycosylation Mix II (O- and N-deglycosylation; New England Biolabs P6044S) or PNGase F (N-deglycosylation; New England Biolabs P0704L). After denaturation, samples were separated by Bolt 4–12% Bis-Tris Plus Gels (Thermo Fisher Scientific NW04125BOX) and transferred onto a PVDF membrane using iBlot Transfer Stacks (Thermo Fisher Scientific IB401001).

Blots were blocked with 5% BLOT-QuickBlocker (G Biosciences 786-011) in TBST for 1 h at room temperature. Blots were then probed with different primary antibodies [phospho- NF-κB p65 Ser536 (Cell Signaling Technology 3033S, 1:1000), NF-κB p65 (Santa Cruz Biotechnology sc-8008, 1:200), phospho-STAT1 Tyr701 (Cell Signaling Technology 9167S, 1:1000), STAT1 (Cell Signaling Technology 9172S, 1:1000), CD276 (R&D Systems AF1027, 1:200), CD70 (Santa Cruz Biotechnology sc-365539, 1:200), CD58 (Thermo Fisher Scientific MA5800, 1:200), NECTIN2 (R&D Systems AF2229, 1:2000), HLA-A (Abcam ab52922, 1:5000), TNFRSF1A (Santa Cruz Biotechnology sc-8436, 1:200), IFNGR2 (R&D Systems AF773, 1:200), FAS (Santa Cruz Biotechnology sc-8009, 1:200), IFNAR1 (Santa Cruz Biotechnology sc-7391, 1:100), TNFRSF10B (Novus Biologicals NB100-56618, 1:200), MICB (R&D Systems MAB1599-100, 1:500), TNFRSF10A (R&D Systems AF347, 1:200), PVR (R&D Systems MAB25301, 1:500), MICA (R&D Systems MAB1300-100, 1:500), HMGB1 (Abcam ab18256, 1:1000), 4-1BBL (TNFSF9; R&D Systems AF2295, 1:200), NT5E (Abcam ab175396, 1:1000), ULBP2 (R&D Systems AF1298, 1:2000), IFNGR1 (R&D Systems MAB6731, 1:500), ULBP3 (R&D Systems AF1517, 1:2000), CD39 (Abcam ab108248, 1:1000), FLAG (Millipore Sigma F7425, 1:1000), or GAPDH (Cell Signaling Technology 2118L, 1:1000)] in 2.5% BLOT-QuickBlocker (G Biosciences 786-011) in TBST overnight at 4 °C. Blots were washed with TBST before incubation with secondary antibodies [Anti-rabbit IgG-HRP (Cell Signaling Technology 7074S, 1:5000), Anti-mouse IgG-HRP (Cell Signaling Technology 7076S, 1:5000), anti-goat IgG-HRP (Santa Cruz Biotechnology sc-2354, 1:5000)] in 2.5% BLOT-QuickBlocker (G Biosciences 786-011) in TBST for 1 h at room temperature. Blots were washed with TBST and imaged using chemiluminescent substrate [Pierce ECL (Thermo Fisher Scientific 32209), SuperSignal West Pico PLUS (Thermo Fisher Scientific 34577), or SuperSignal West Femto (Thermo Fisher Scientific 34096)] on the ChemiDox XRS + (Bio-Rad).

**Flow cytometry assays**. Per condition, $5 \times 10^5$ cells were pelleted at $200 \times g$ for 5 min and washed once with PBS. Cell were fixed in 4% formaldehyde in PBS at 4 °C for 10 min. Cells were washed twice with PBS and resuspended in PBS with 25 μg/mL recombinant Fc chimera proteins [PVRIG (R&D Systems 9365-PV-050),

CD226 (R&D Systems 666-DN-050), NKG2D (R&D 1299-NK-050), TREML2 (R&D Systems 3259-TL-050), CD2 (R&D Systems 1856-CD-050), CD96 (R&D Systems 9360-CD-050), TIGIT (BPS Bioscience 71186), CD27 (BPS Bioscience 71176), or 4-1BB (TNFRSF9; Sino Bio 10041-H03H)], 0.1 μg/mL HLA-A2:NY-ESO-1 Fab[65], 5 μg/mL Fas antibody (Millipore Sigma 05-201), 25 μg/mL TNFRSF10B antibody (Novus Biologicals NB100-56618, 1:200), or Dylight 649 labeled Tomato Lectin (Vector Laboratories DL-1178, 1:100). Cells were incubated at 4 °C for 1 h. Cells were washed twice with PBS and resuspended in PBS with the appropriate secondary antibody [IgG Fc PE (Thermo Fisher Scientific 12-4998-82, 1:50), His Tag Alexa Fluor 647 (Thermo Fisher Scientific MA121315A647, 1:500), mouse Alexa Fluor 568 (Thermo Fisher Scientific A-11031, 1:400), or rabbit Alexa Fluor 647 (Thermo Fisher Scientific A-21244, 1:400)]. Cells stained with Tomato Lectin were not incubated with additional secondary antibodies. Cells were incubated at 4 °C for 30 min. Cells were washed twice with PBS. For each sample, 10,000 cells were analyzed on a CytoFLEX Flow Cytometer (Beckman Coulter) and quantified with FlowJo 10.8.1. For each experiment, median fluorescence intensities for three biological replicates were compared to determine statistical significance.

**Primary patient-derived cell models.** CCLF_MELM_0011_T melanoma tumor tissue and CCLF_PANC_0014_T pancreatic tumor tissue were obtained from Dana-Farber Cancer Institute hospital with informed consent and the cancer cell model line generation was approved by the ethical committee. Both tumor tissues were freshly received into the lab, rinsed with 95–100% ethanol very quickly, and 1× PBS twice. Tissue was transferred to a sterile Petri dish and the tissue was minced into small 1–2 mm fragments. Dissected tissues were dissociated in a collagenase/hyaluronidase (StemCell technologies 07912) medium for 1 h. The red blood cells were further depleted by adding Ammonium Chloride Solution (StemCell technologies 07800). CCLF_MELM_0011_T dissociated cells were plated with smooth muscle growing medium-2 (Lonza CC-3181) into a six-well plate, media was changed every 2–3 days, and cells were split when confluency of 80% was reached. A 1:3 ratio was used when splitting CCLF_MELM_0011_T. CCLF_PANC_0014_T dissociated cells were plated into a 24-well plate with a 50:50 mix of Clevers pancreas organoid media[66]: Propagenix Conditioned media (Propagenix 256-100) and split when confluency of 80% was reached. Media was changed every 3–4 days. A 1:2 ratio was used when splitting CCLF_PANC_0014_T which is a mixed population of suspension and adherent cells. Both lines were passaged five times before a pellet was taken for sequencing verification. The confirmed melanoma cell model and confirmed pancreatic adenocarcinoma cell model were propagated for another 10–15 passages and their cryovials preserved. CCLF_MELM_0011_T passage 11 cells and CCLF_PANC_0014_T passage 20 cells were used for this study.

**Statistics.** Statistical tests were applied with the sample size listed in the text and figure legends. The sample size represents the number of independent biological replicates. Data supporting main conclusions represent results from at least two independent experiments. All graphs with error bars report mean ± s.e.m. values. PRISM was used for basic statistical analysis and plotting (http://www.graphpad.com), and the R language and programming environment (https://www.r-project.org) was used for the remainder of the statistical analysis.

**Reporting summary.** Further information on research design is available in the Nature Research Reporting Summary linked to this article.

## Data availability

The sequencing data generated in this study has been deposited in the Gene Expression Omnibus under accession code GSE159540. The mass spectrometry proteomics data have been deposited to the ProteomeXchange Consortium via the PRIDE partner repository with the accession code PXD031532. The Cancer Genome Atlas datasets were downloaded from the Broad GDAC Firehose (http://gdac.broadinstitute.org/) using the TCGA2STAT package for R[56]. The human hg38 genome was downloaded from the UCSC Genome Browser (https://genome.ucsc.edu/). The remaining data are available within the Article, Supplementary Information, or Source Data file. Source data are provided with this paper.

## Code availability

Code for the analyses described in this study is available on Github (https://github.com/fengzhanglab/Joung_Immunotherapy_Manuscript)[67].

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

## Acknowledgements

We thank the Broad Institute Vivarium staff for NSG mice support; Eric Spooner of Whitehead Institute for mass spectrometry; Kai Wucherpfennig for critical reading of the manuscript; Andrew Tang for assistance with the illustration of Figs. 3g and 4e; Richard Belliveau for overall research support; and the entire Zhang laboratory for support and advice. J.J. is supported by a NIH F31 fellowship 1F31-MH117886. S.P.N. is supported by the Paul & Daisy Soros Fellowship. F.Z. is supported by NIH grants 2R01-HG009761 and 1DP1-HL141201, the Howard Hughes Medical Institute, Open Philanthropy, the Edward Mallinckrodt Jr. Foundation, the Poitras Center for Psychiatric Disorders Research at MIT, the Hock E. Tan and K. Lisa Yang Center for Autism Research at MIT, the Yang-Tan Center for Molecular Therapeutics at MIT, and the Phillips family, R. Metcalfe, and J. and P. Poitras.

## Author contributions

J.J. and F.Z. conceived and designed the study. J.J., P.C.K., A.S., and S.P.N. performed experiments. J.J. analyzed the data. J.H.C. helped establish the tumor xenograft models for in vivo validation and performed T cell intravenous injections. R.C.L. advised on establishing T cell cytotoxicity assays and provided T cells under the supervision of M.V.M. R.D. and Y.Y.T. acquired and generated the primary patient-derived tumor cell models. F.Z. supervised the research and experimental design with support from R.M. J.J., R.M., and F.Z. wrote the paper with help from all authors.

## Competing interests

J.J. and F.Z. are inventors listed on a U.S. Provisional Patent Application No. 63/196,520 entitled "Novel Targets for Enhancing Anti-Tumor Immunity". F.Z. is a cofounder of Editas Medicine, Beam Therapeutics, Pairwise Plants, Arbor Biotechnologies, and Sherlock Biosciences. M.V.M. is on the scientific advisory board of Cabaletta Bio, Cel-lectis, In8bio, TCR2, and WindMIL, as well as the board of directors of 2Seventy Bio. M.V.M. is an equity holder of Century Therapeutics, Genocea, Oncternal, TCR2, and 2Seventy Bio. The remaining authors declare no competing interests.
