## [Peer Review File · Nature Communications]

Reviewers' Comments:

Reviewer #4:

Remarks to the Author:

In the manuscript by Joung et al the authors perform a genome-scale CRISPR activation screen to identify targets that install immune evasion in melanoma tumor cells. From this screen they identify 4 major hits that are further investigated in the described studies. These targets include: CD274, MCL1, JUNB and B3GNT2. As a glyco-immunology expert, I was asked by Nature Communications to particularly address the glyco-immunological aspects of the work, so that is what I will focus on below.

In my opinion the authors make quite a compelling case that B3GNT2 is involved in immune evasion, however the functional validation is quite meager. The authors identify their targets just based on an increase in apparent molecular weight. Although this can indeed be due to the action of B3GNT2, so can by no means be sure of this, just by doing a western blot for protein size. Glycosylation is not a linear process but a complex interplay of many enzymes working as an assembly line in the Golgi. Deleting one enzyme or overexpressing one enzyme disrupt the whole assembly line and could therefore result in many different glycosylation alterations, which may not be directly coming from the overexpressed glycosyltransferase.

Along that same line, kifunensine disrupts the ENTIRE N-glycosylation pathway, changing ALL N-glycans to a high mannose-type. Thus, the reversal of the phenotype might not be due to the reversal of the B3GNT2 action, but could be an entirely different mechanism or could even be due to different target proteins, which act differently when carrying high mannose.

Thus, a more detailed analysis of the glycosylation of the target proteins is absolutely necessary. I do wonder why the authors haven't employed their Tomato lectin, which works perfectly well on Western Blots or in IPs. Therefore, I urge the authors to perform one the following experiments:

- 1) Probe the blot from Fig 4B and extended data figure 8C & D with Tomato lectin to prove that the target protein indeed now carry extended LacNAc structures
- 2) Perform an IP with Tomato lectin on the control and B3GNT2-modified cells and blot with antibodies for the respective protein targets.

Finally, I was wondering why only B3GNT2 would have this immune evasive effect. B3GNT2 belongs to a group of 8 similar B3GNT enzymes with highly redundant functions. What is special about B3GNT2? I would appreciate if the authors could comment on this in their manuscript.

Response to Reviews for NCOMMS-21-44860

CRISPR activation screen identifies BCL-2 proteins and *B3GNT2* as drivers of immunotherapy resistance

Joung *et al.*

We thank the Reviewer for the positive and helpful feedback. Based on this feedback, we have revised the manuscript to include additional experimental evidence verifying that B3GNT2 adds poly-LacNAc to 10 ligands and receptors as detailed below.

Referee #4

In the manuscript by Joung et al the authors perform a genome-scale CRISPR activation screen to identify targets that install immune evasion in melanoma tumor cells. From this screen they identify 4 major hits that are further investigated in the described studies. These targets include: CD274, MCL1, JUNB and B3GNT2. As a glyco-immunology expert, I was asked by Nature Communications to particularly address the glyco-immunological aspects of the work, so that is what I will focus on below.

In my opinion the authors make quite a compelling case that B3GNT2 is involved in immune evasion, however the functional validation is quite meager. The authors identify their targets just based on an increase in apparent molecular weight. Although this can indeed be due to the action of B3GNT2, so can by no means be sure of this, just by doing a western blot for protein size. Glycosylation is not a linear process but a complex interplay of many enzymes working as an assembly line in the Golgi. Deleting one enzyme or overexpressing one enzyme disrupt the whole assembly line and could therefore result in many different glycosylation alterations, which may not be directly coming from the overexpressed glycosyltransferase.

Along that same line, kifunensine disrupts the ENTIRE N-glycosylation pathway, changing ALL N-glycans to a high mannose-type. Thus, the reversal of the phenotype might not be due to the reversal of the B3GNT2 action, but could be an entirely different mechanism or could even be due to different target proteins, which act differently when carrying high mannose.

Thus, a more detailed analysis of the glycosylation of the target proteins is absolutely necessary. I do wonder why the authors haven't employed their Tomato lectin, which works perfectly well on Western Blots or in IPs. Therefore, I urge the authors to perform one the following experiments:

- 1) Probe the blot from Fig 4B and extended data figure 8C & D with Tomato lectin to prove that the target protein indeed now carry extended LacNAc structures

2) Perform an IP with Tomato lectin on the control and B3GNT2-modified cells and blot with antibodies for the respective protein targets.

We thank the Reviewer for this thoughtful suggestion. We chose to perform IP using tomato lectin on A375 cells overexpressing GFP control or *B3GNT2* followed by Western blot (option 2). Our results show that at baseline, all 10 ligands and receptors had some poly-LacNAc modifications, which increased in proportion and length upon overexpression of *B3GNT2* (Fig. 4b).

We did try probing the blots from Fig. 4b and ED Fig. 8c,d with tomato lectin (option 1). However, we found that the results were inconclusive because B3GNT2 has at least 10 targets with overlapping size distributions, which made it difficult to assign tomato lectin binding to respective targets.

Finally, I was wondering why only B3GNT2 would have this immune evasive effect. B3GNT2 belongs to a group of 8 similar B3GNT enzymes with highly redundant functions. What is special about B3GNT2? I would appreciate if the authors could comment on this in their manuscript.

We have added discussion of B3GNT2 compared to other B3GNT enzymes. Although there are other B3GNT family members with overlapping function, we did not identify any other B3GNT enzymes in our screen. This may be because B3GNT2 has the strongest poly-LacNAc synthesis activity in vitro relative to other B3GNT enzymes and is therefore considered the main poly-N-acetyllactosamine (poly-LacNAc) synthase (PMIDs: 20816167, 17113861).

Reviewers' Comments:

Reviewer #4:

Remarks to the Author:

I thank the authors for their revisions and answers to my comments.

Response to Reviews for NCOMMS-21-44860A

CRISPR activation screen identifies BCL-2 proteins and B3GNT2 as drivers of cancer resistance to T cell-mediated cytotoxicity

Joung *et al.*

Referee #4

I thank the authors for their revisions and answers to my comments.

 | We thank the Reviewer for the positive and helpful feedback, which improved our
 | manuscript.